# AAAKB: A manually curated database for tracking and predicting genes of Abdominal aortic aneurysm (AAA)

Di Yao[1], Shuyuan Mei[2], Wangyang Tang[3], Xingyu Xu[3], Qiulun Lu[2]*, Zhiguang Shi[2]*

**1** Institute of Industrial Internet and Internet of Things, China Academy of Information and Communications Technology (CAICT), China, **2** Key Laboratory of Cardiovascular and Cerebrovascular Medicine, School of Pharmacy, Nanjing Medical University, Nanjing, China, **3** School of Biomedical Engineering and Informatics, Nanjing Medical University, Nanjing, China

☯ These authors contributed equally to this work.
\* shizg_bioinfo@163.com (ZS); Qiulunlu@njmu.edu.cn (QL)

**Data Availability Statement:** The gene information manually collected from literature and AAA related genes predicted by random forest model can be obtained from AAAKB website (http://www.lqlgroup.cn:3838/AAAKB/). Other relevant data are

## Abstract

Abdominal aortic aneurysm (AAA), an extremely dangerous vascular disease with high mortality, causes massive internal bleeding due to aneurysm rupture. To boost the research on AAA, efforts should be taken to organize and link the information about AAA-related genes and their functions. Currently, most researchers screen through genetic databases manually, which is cumbersome and time-consuming. Here, we developed "AAAKB" a manually curated knowledgebase containing genes, SNPs and pathways associated with AAA. In order to facilitate researchers to further explore the mechanism network of AAA, AAAKB provides predicted genes that are potentially associated with AAA. The prediction is based on the protein interaction information of genes collected in the database, and the random forest algorithm (RF) is used to build the prediction model. Some of these predicted genes are differentially expressed in patients with AAA, and some have been reported to play a role in other cardiovascular diseases, illustrating the utility of the knowledgebase in predicting novel genes. Also, AAAKB integrates a protein interaction visualization tool to quickly determine the shortest paths between target proteins. As the first knowledgebase to provide a comprehensive catalog of AAA-related genes, AAAKB will be an ideal research platform for AAA. Database URL: http://www.lqlgroup.cn:3838/AAAKB/.

## Introduction

Abdominal aortic aneurysm (AAA), a potentially life-threatening condition, is characterized as a weak spot in the blood vessel wall. Its rupture causes high mortality if not treated [1]. AAA is commonly found in adult patients, especially elderly men, and leads to severe complications [2]. Multiple factors, including genetic interactions, environmental influences and lifestyle, contribute to the manifestations of AAA. This complex vascular disease is regulated by multiple genes. Currently, there is no effective drug or therapy to treat or prevent AAA [3–5]. To address this issue, targeting specific genes could be a potential strategy to treat AAA. Thus,

within the manuscript and its Supporting Information files.

**Funding:** This study was supported by the grant from the National Natural Science Foundation of China (81970414 to Q. L). This work was supported by the 2018 Industrial Internet Innovation and Development Project —Industrial Internet Identification Resolution System: National Top-Level Node Construction Project (Phase I). The funders had no role in study design, data collection and analysis, decision to publish, or preparation of the manuscript.

**Competing interests:** The authors have declared that no competing interests exist.

understanding the mechanism of AAA progression requires the exploration of known and novel genes.

As a highly studied disease, scattered information of AAA was buried in the piles of literature, which makes it difficult for researchers to clarify genes, molecular complexes, and signaling pathways involved in AAA. This consumes tremendous amount of time for researchers to sort out related proteins and genes they are interested in. What is more, complicated by genetic modification, identifying novel AAA genes remains a challenge, especially in mammalian models [6–8]. Manually digging up AAA-related genes and intertwining signaling pathways may result in missing information hence leads to inconclusive results. An electronic database is required to integrate information from disparate sources and predict potential candidate genes associated with AAA.

Besides, protein-protein interaction (PPI) has increasingly been used in bioinformatics studies to predict disease-related genes [9–11]. Recently, machine learning methods for analyzing PPI data have become popular and have been applied to different aspects of biological studies, including gene classification and functional prediction [12, 13]. Some methods used to predict disease genes are based on constructing combinatorial classifiers using combinatorial PPI network topological features. For example, Han et al. put forward a novel framework to identify the multiple sclerosis-related genes [14]. The framework captured the topological structure of the PPI network and trained the SVM classifier to predict disease-related genes. The strong calculating capacity of machine learning methods could help to address the complexity of AAA. SVM was applied to sifting complex genetic regulation in AAA.

Here, we developed a comprehensive database of abdominal aortic aneurysm, known as AAAKB. It is the first manually curated knowledgebase with detailed annotation of genes associated with AAA based on literature references and reviewed databases. AAAKB has documented 378 genes identified up-to-date through various experimental methods such as RT-PCR, western blotting, immunochemistry, and cell-based assays. Besides, the AAA genes have been associated with gene ontology terms, pathways, and SNPs. In addition, it predicted genes that may function in AAA by analyzing PPI networks with RF. This approach has enabled us to identify new genes and novel pathways associated with AAA. We have also equipped the database with a tool to visualize the shortest interaction path between selected proteins, allowing researchers to quickly understand the potential molecular mechanisms of new genes in AAA. By providing this information, AAAKB facilitates efficient interrogation of published data on AAA, allowing researchers and clinicians to identify gaps in our current understanding of AAA and its associated comorbidities. Nevertheless, while PPI-based predictions of new AAA genes have their benefits, they can be one-dimensional, and the information provided by AAAKB may be limited. Consequently, our forthcoming work will focus on integrating various omics data to generate more trustworthy prediction outcomes and expand the database's content.

## Materials and methods

### Manual curation of literature

PubMed, was searched using relevant keywords namely 'abdominal aortic aneurysm [Title/Abstract]'. The genes cited in the retrieved literature were obtained from the NCBI gene database. References were reviewed to determine the association of each of these genes with AAA.

### Construction of positive and negative gene sets

During the construction of the positive and negative gene sets, we handled human genes and mouse genes separately. Positive genes were defined as all AAA genes collected from the

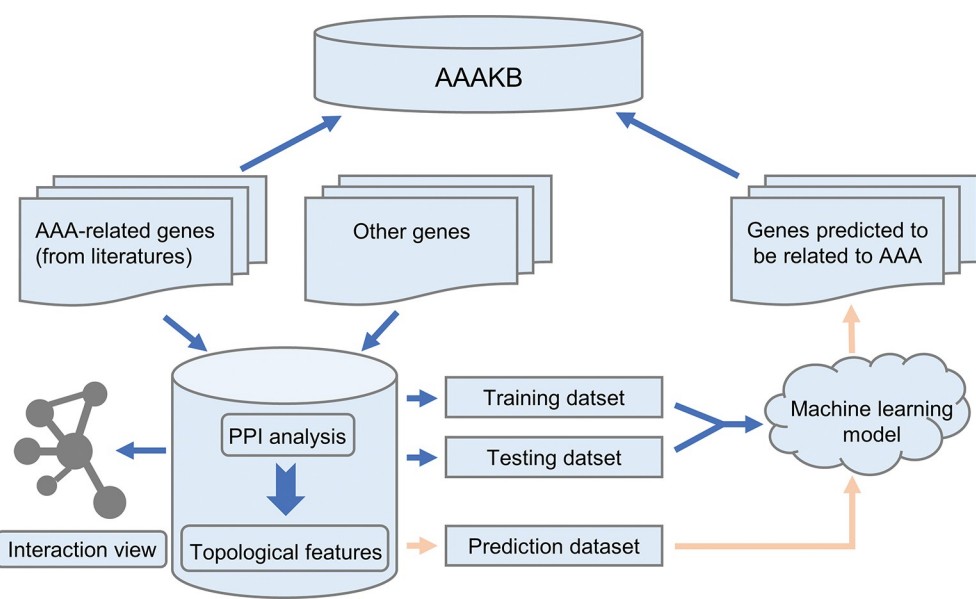

**Fig 1. Abdominal aortic aneurysm knowledgebase (AAAKB) scheme.** Manually curated AAA-related genes were collected and further used to predict AAA candidate genes.

literature. The negative genes consisted of genes randomly selected from the protein-protein interaction network that were not present in the positive gene set (Fig 1). We hypothesized that the positive gene set is unrelated to AAA genes.

## PPI network topological features

The protein interaction data was downloaded from the Biogrid database. We computed seven commonly-used measurements for each gene in the positive and negative gene sets:

1. Degree (D): The number of edges connected to a node.

2. Neighbor count of disease genes (N): The number of disease genes connected to a node.

3. Ratio of disease genes in neighborhood (R): The proportion of a node's neighbors that are disease genes.

4. Betweenness centrality (B): The number of shortest paths between other nodes that run through the node of interest.

5. Clustering coefficient (C): The proportion of a node's neighbors that are also connected to each other.

6. Mean shortest path length to disease gene (M): The average shortest path length between a node and all the disease genes in the network.

7. PageRank centrality (PR): A measurement of the importance of a node in a network, based on the idea that a node is important if it is linked to by other important nodes[Reprint of: The anatomy of a large-scale hypertextual web search engine - ScienceDirect].

## Establishment and evaluation of machine learning models

After mixing the positive and negative gene sets, we divided the data into a training set of 70% and a test set of 30%. We applied four machine learning algorithms, including support vector

machines (SVM), naive Bayes (NB), random forest (RF), and k-nearest neighbors (KNN), to generate classifiers using the training data sets. We set a seed of '123' for reproducibility. To implement the SVM and NB models, we used the R package 'e1071'. We set the ranges of two hyperparameters, 'gamma' and 'cost', for tuning the SVM model using cross-validation. The NB model runs according to default parameters. We implemented a random forest classification model using the 'randomForest' package in R. This model is tuned using the hyperparameters 'mtry' and 'ntree'. The kNN model is implemented using the R package 'class' and tuned by the hyperparameter 'k'. We performed 10-fold cross-validation on the training data to determine the best hyperparameters for all models.

To obtain a robust estimate of the model's accuracy, we repeated all four models 200 times with randomly sampled training and testing sets. We evaluated the performance of these machine learning models using several metrics, including accuracy, precision, recall, and F1 score. For each randomization, we tested whether each gene could be classified as a disease gene. If so, we included this gene as a candidate gene in our final results.

### Differential gene expression analysis

The gene expression profile of 29 patients and 10 control aortic specimens was obtained from the GEO dataset: GSE57691. The limma package [15] was used to identify the differentially expressed genes. The list of differentially expressed genes of AAA mouse model was provided by the study of Boytard L et al. [16]. A fold change >1.5 and an adjusted p-value < 0.05 were used as screening criteria for identifying the differentially expressed genes. Functional enrichment analysis was performed by Metascape [17].

### Implementation

AAAKB has been implemented in R as a Shiny application. The R package 'cisPath' was integrated into the website to display the PPI networks between a pair of given proteins.

## Results

### The manual curation of AAA-related genes from literature

After removing deduplication, AAAKB contains information of 378 genes and 434 SNPs (Table 1), which have been validated to be AAA associated. According to the reported main research vectors, the genes were divided into two species groups of human and mouse. General information is provided, including gene ID, basic description, genomic location, references and functional annotation. As described before, AAA is regulated by inflammatory response, oxidative stress, extracellular matrix degradation, and apoptosis. Genes in AAAKB showed significant enrichment of GO terms, such as 'inflammatory response', 'response to decreased oxygen levels', 'extracellular matrix' and 'negative regulation of cell population proliferation'. Our results of enrichment analysis are consistent with previous studies (Fig 2). Moreover, the

**Table 1. Number of entries in the datasets of AAAKB.**

| Dataset | Entries |
|---|---|
| AAA-related genes | Human: 213; Mouse: 165 |
| AAA-related SNPs | Human:434 |
| Gene ontologys | Human: 7398; Mouse: 7144 |
| Pathways | Human: 241; Mouse: 218 |
| Candidates | Human: 205 |

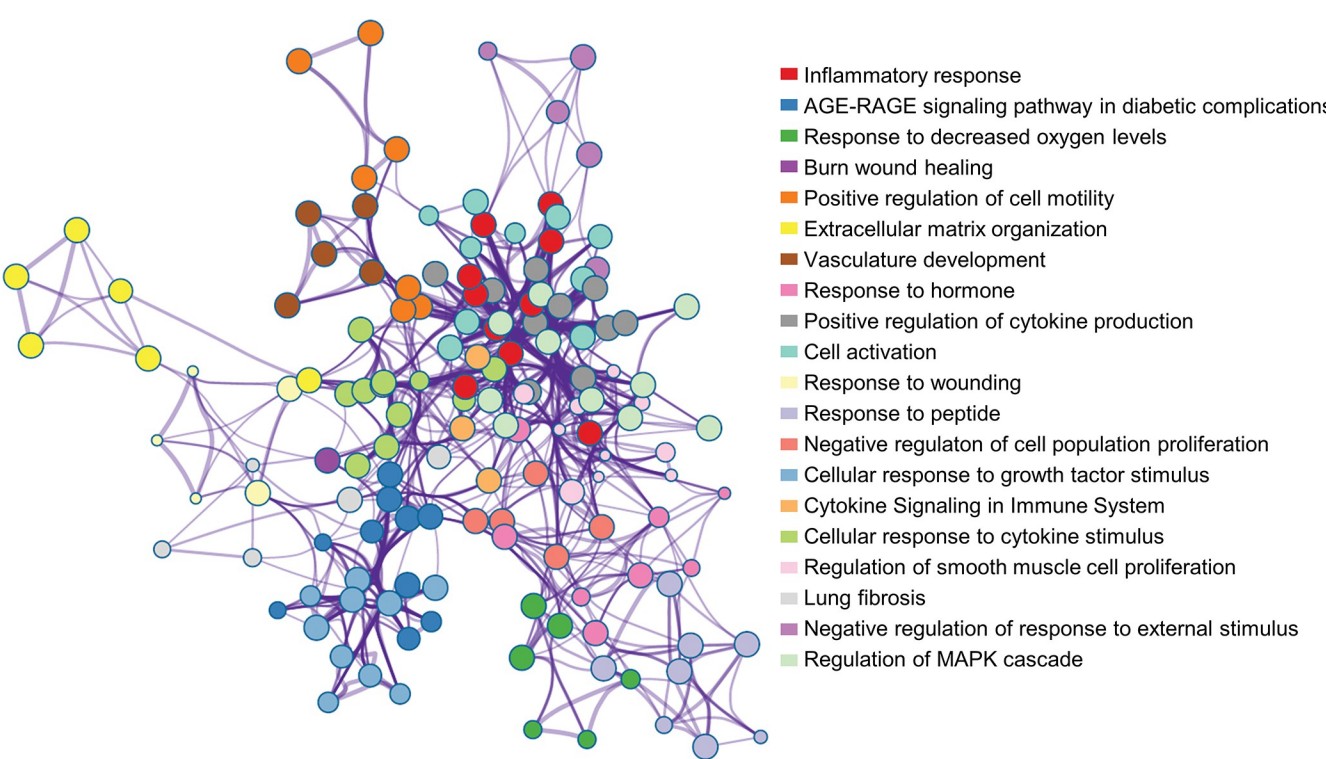

**Legend:**
- ■ Inflammatory response
- ■ AGE-RAGE signaling pathway in diabetic complications
- ■ Response to decreased oxygen levels
- ■ Burn wound healing
- ■ Positive regulation of cell motility
- ■ Extracellular matrix organization
- ■ Vasculature development
- ■ Response to hormone
- ■ Positive regulation of cytokine production
- ■ Cell activation
- ■ Response to wounding
- ■ Response to peptide
- ■ Negative regulaton of cell population proliferation
- ■ Cellular response to growth tactor stimulus
- ■ Cytokine Signaling in Immune System
- ■ Cellular response to cytokine stimulus
- ■ Regulation of smooth muscle cell proliferation
- ■ Lung fibrosis
- ■ Negative regulation of response to external stimulus
- ■ Regulation of MAPK cascade

**Fig 2. Functional enrichment analysis for known genes in AAAKB.** The network of enriched terms is colored by cluster ID, and nodes sharing the same cluster are typically close to each other.

relationship between aortic aneurysms and diabetes is also evident, which is in accord with clinical observation. The vast majority of epidemiological studies have identified an inverse association between diabetes mellitus (DM) and the prevalence and incidence of AAA [18]. Notably, DM leads to an increase in advanced glycation end products (AGEs). AGEs activate receptor for advanced glycation end products (RAGE) and reduce the incidence of AAA. Besides, nearly one-sixth to one-fifth of the genes in AAAKB are transcription factors (S1 Fig), indicating their extensive regulatory roles. These observations reinstate the complex patho-physiology of AAA.

## The overall framework of AAAKB

AAAKB is designed in a user-friendly manner with five main menus, i.e. Home, Browse, Tools, Feedback and Help (Fig 3). The Home page contains a brief introduction to AAA and will navigate the users to the respective pages. Examples that assist users to use the website are displayed on the Help page. To avoid the possibility that some genes might be missing from the data we collected, the Feedback page is included. The Browse page is designed to facilitate efficient searches. The four special search options are described below:

a. Genes: this page provides descriptions of the genes, their genomic locations and links to the complete gene record.

b. GO terms: contains ontology terms associated with AAA-related genes

c. Pathways: contains pathways involving AAA-related genes hyperlinked to the KEGG pathway database.

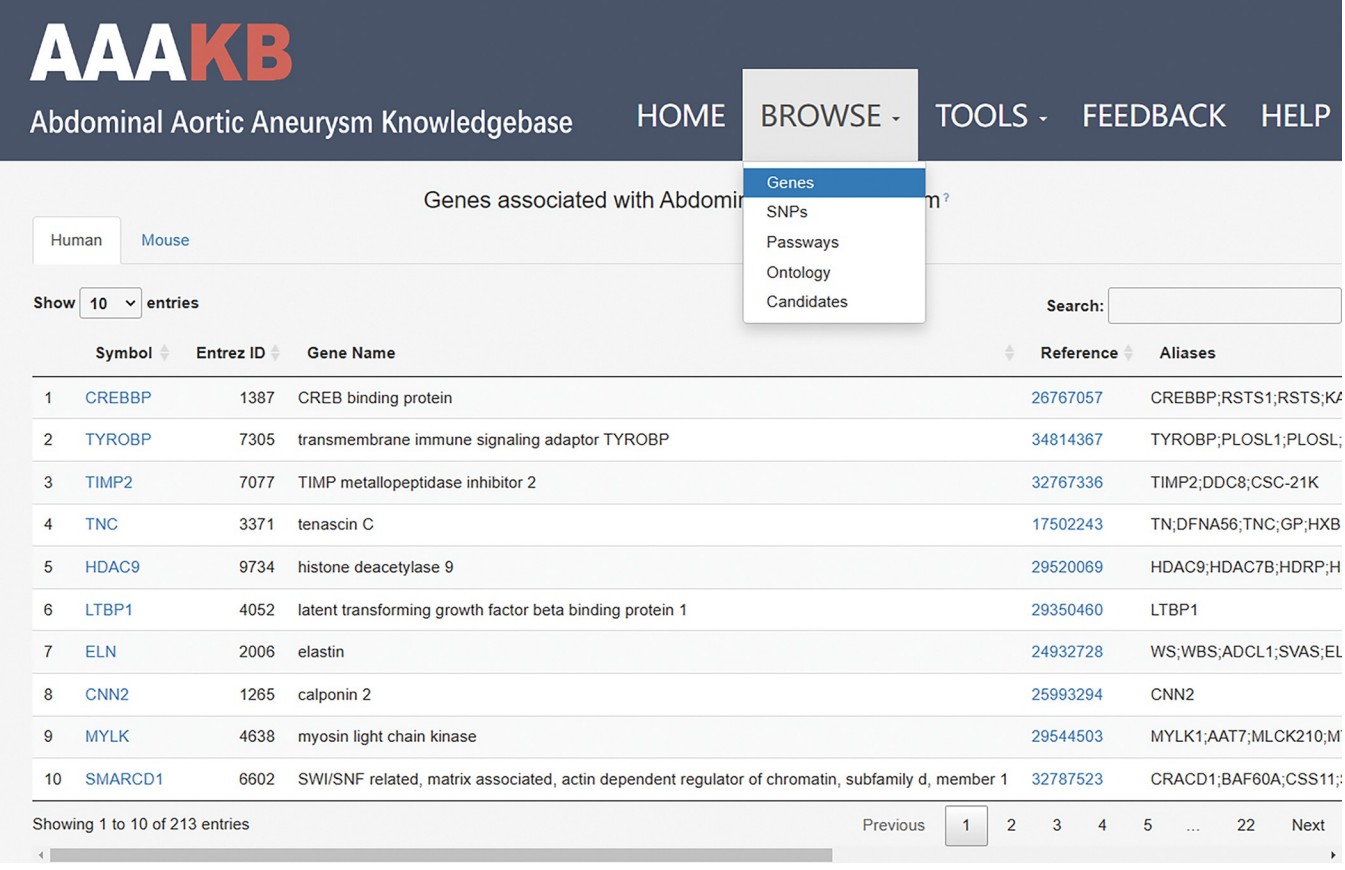

**Fig 3. The browse page of AAAKB.** It provides access to information on genes, SNPs, pathways, gene ontologies and candidates.

d. Candidates: genes predicted to function in AAA by analyzing PPI networks with RF.

 The Tools section enables users to display the PPI networks between a pair of given proteins they are interested in. It provides great convenience and help for users to understand the most direct paths of interactions between AAA-related genes.

## Predicting novel candidate genes for AAA

In general, functionally related genes were clustered in the same network [19–21]. Analysis shows that genes related to specific diseases tend to have higher and more synchronous expression and tend to interact with each other [22]. Therefore, as described in the method, the network topology features were calculated for each gene based on the PPI network. We constructed four machine learning methods for human and mouse data, including SVM, NB, RF, and KNN. The genes collected in our database, as the positive set, provide proofreading of classification to distinguish AAA-related and non-related genes. RF and SVM have better accuracy, approximately 70%, than NB and KNN. Another popular measure, F1 score, recall and AUC, was also calculated which came out to be highest for RF (Fig 4). Therefore, we chose RF as the optimal and final prediction model for predicting all potential AAA genes in the PPI network. By using the RF classifier, 1809 human genes and 312 mouse genes were finally returned. The genes collected in our database provide sufficient information to train good models.

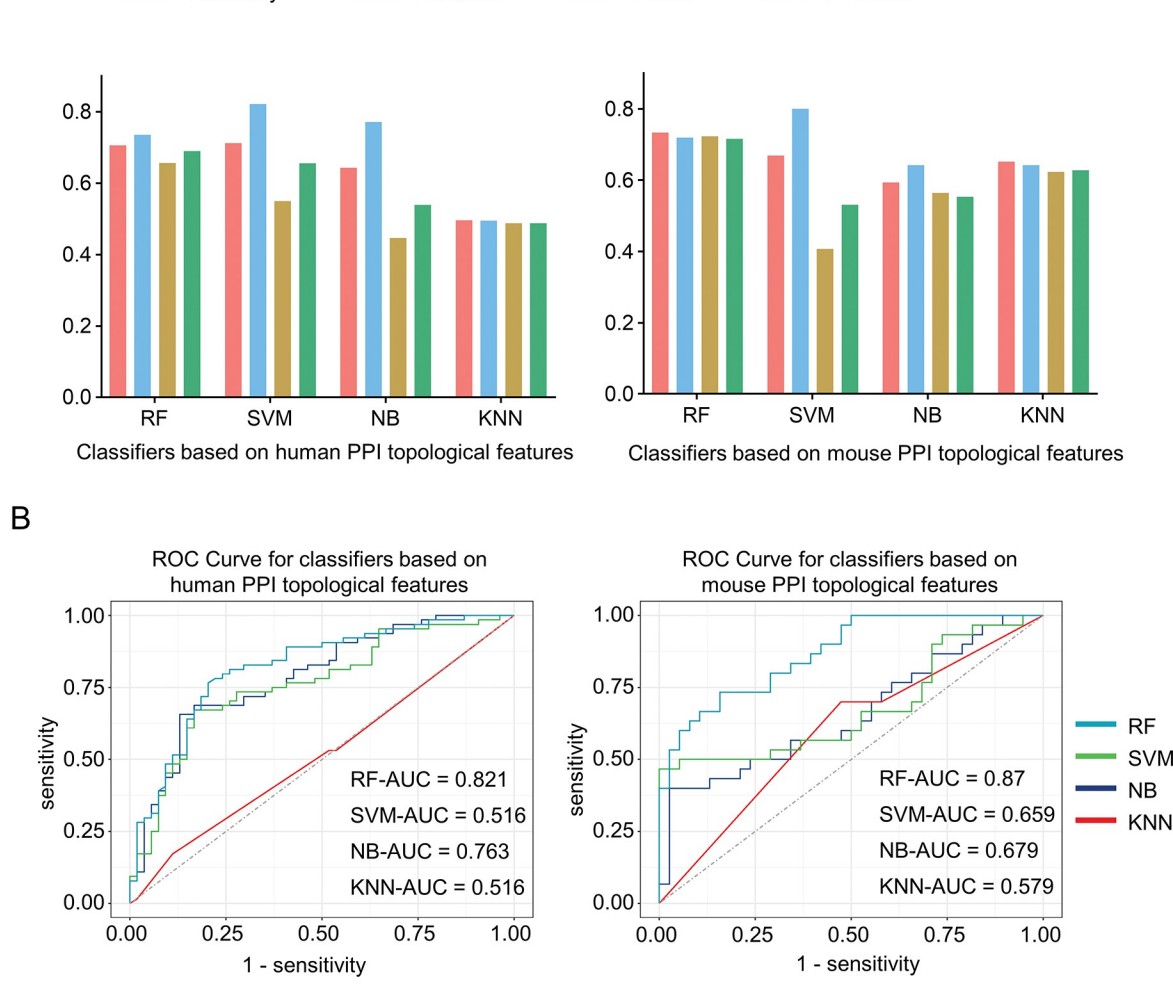

**Fig 4. Performance measures of different classifiers for predicting human and mouse candidate genes.** (A) Shows the evaluation of the four classifiers using the performance measures of Accuracy, Precision, Recall, and F1 score. (B) Shows the ROC curve of the four classifiers.

## Gene expression analysis of candidate genes

To investigate the association between candidate genes and diseases, we assessed the gene expression levels in aortic aneurysm patients and aortic aneurysm mouse models. Using the same screening threshold, we identified 2170 and 312 differentially expressed genes in human and mouse samples, respectively (Fig 5A, S1 Table). Interestingly, only 13% of predicted AAA-related genes in humans were found to be differentially expressed, while in mice, this was only 2.5% (Fig 5B). Our findings suggest that further research should focus on the role of these differentially expressed genes in the development of AAA, as they are enriched in inflammatory reactions, cell death, cell migration, and other biological processes relevant to the pathological progress of AAA (Fig 5C). Additionally, some of the predicted genes, including BCL6 [23], LDB3 [24], MUC1 [25] and SYNC [26], have been implicated in other cardiovascular diseases. However, it is important to acknowledge that factors other than gene expression, such as protein modification [27, 28], also play a crucial role in regulating pathological processes and should not be overlooked.

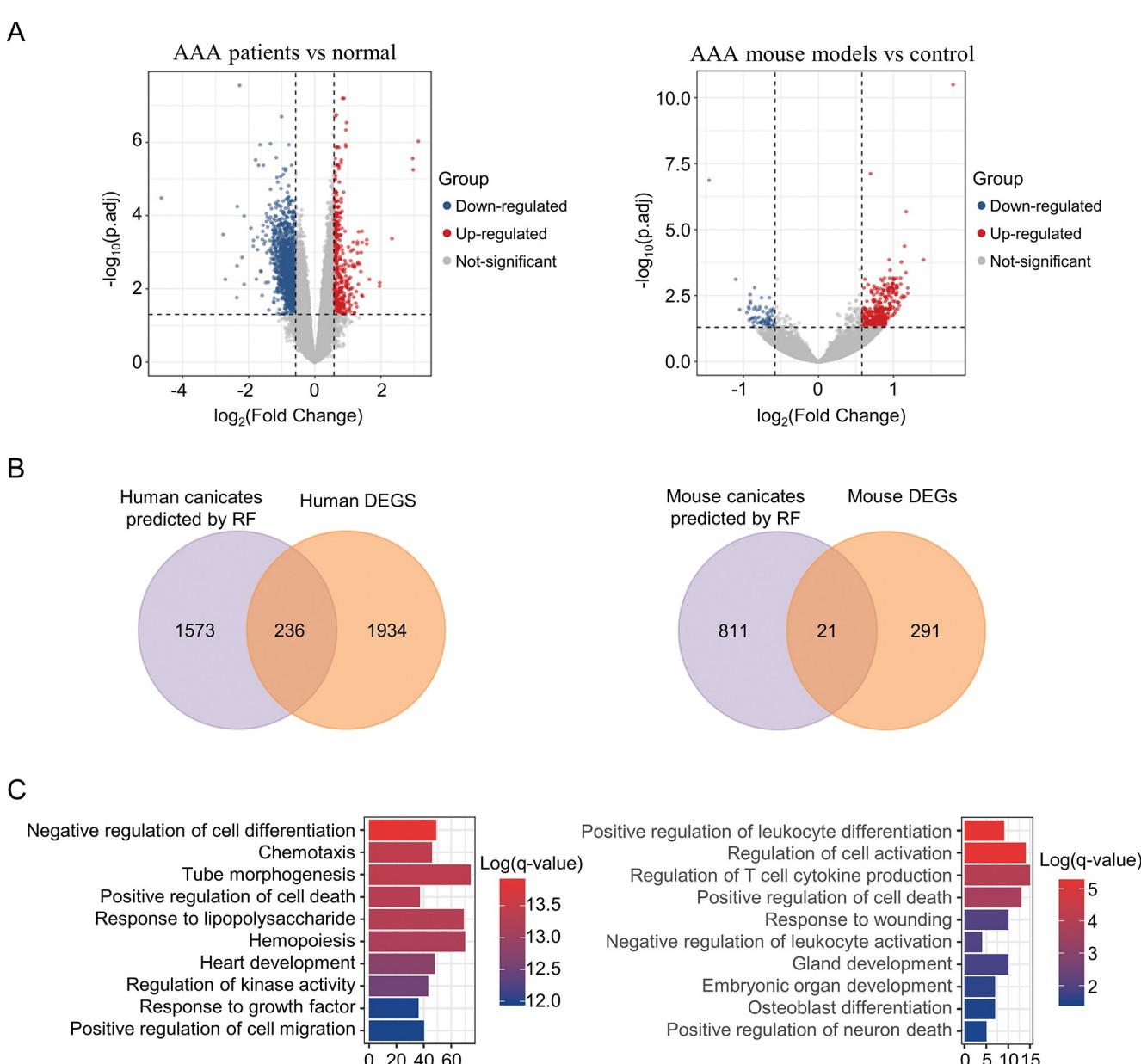

**Fig 5. Separate gene expression analysis in AAA patients and mouse models compared to normal controls.** (A) Volcano plots of differentially expressed genes. (B) Intersections of differentially expressed genes and candidate genes predicted by RF. (C) Gene ontology analysis of genes in the intersections.

## Visualization of PPI networks between a pair of given proteins

PPI networks are often used to study cellular function, disease mechanisms, and drug design. When doing mechanism research, researchers often struggle to figure out the complete upstream and downstream networks of their target genes. To fulfill this need, we equipped the web page with 'cisPath', an R package for visualization and management of functional protein interaction networks [29]. With this package, network diagrams can be derived from one or several given proteins based on PPI data downloaded from the STRING database. Additionally, the package can also identify the shortest interaction path between a pair of given proteins (Fig 6). However, running this package requires some programming ability that is not friendly

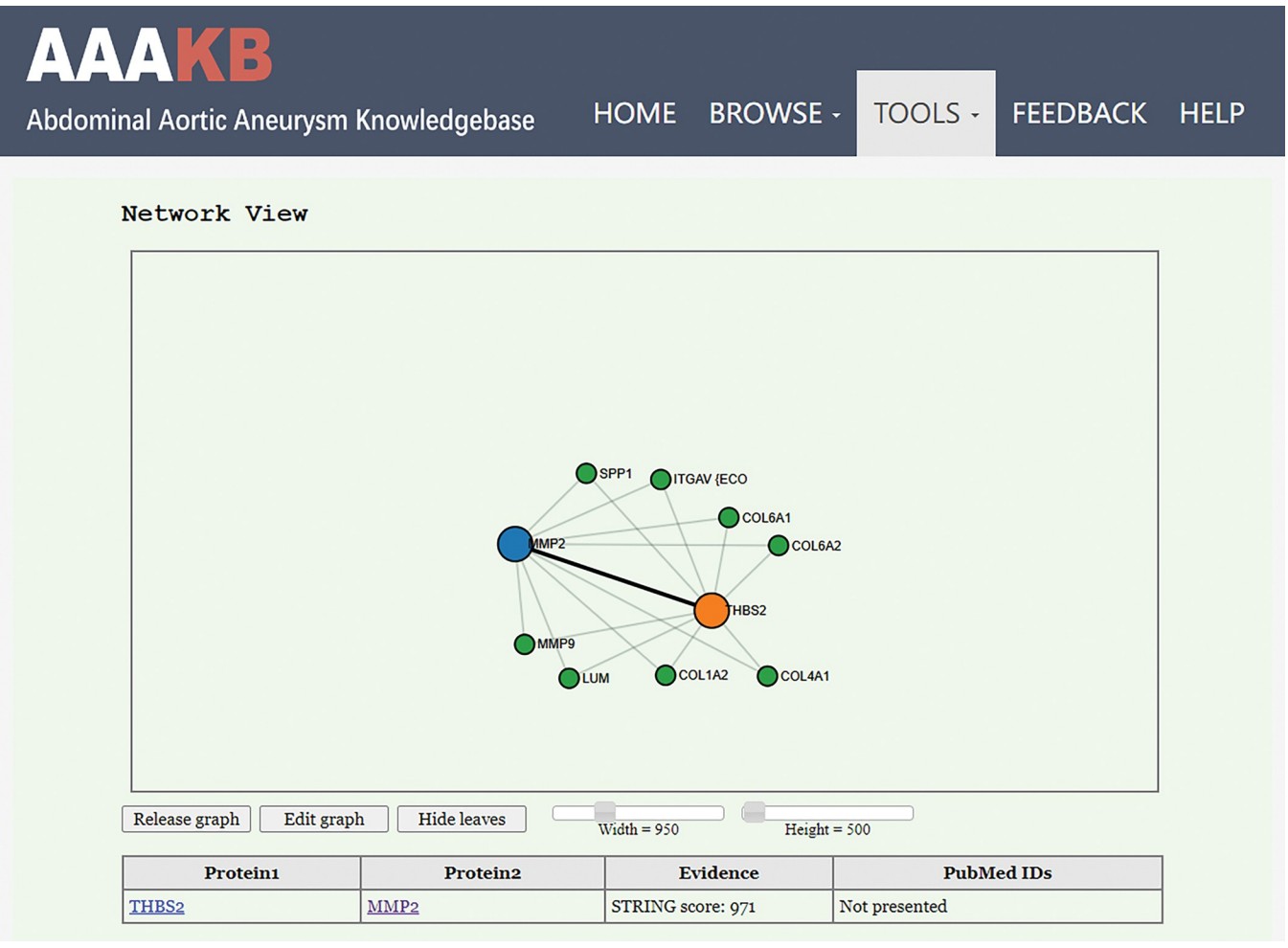

**Fig 6. The shortest interaction path between a pair of given proteins.** AAAKB was equipped with R package 'cisPath'. The shortest interaction path was derived from two given proteins based on PPI data downloaded from the STRING database.

to some molecular biologists. By deploying the R package to our web page, researchers can run cisPath with the click of a mouse and easily view the results through a web browser.

## Discussion

Even though numerous studies have revealed abundant critical regulators involved in AAA, the information for these genes is scattered among piles of papers. It is difficult to gather genetic information and figure out their relationships from the astronomical number of studies. Therefore, a specialist platform is required to assist scientists to understand the function and interactions of these genes quickly and efficiently.

In general, a knowledgebase is a centralized repository of information. Constructing such knowledgebase has become a widely accepted tool in biological studies. For instance, the Reactome Knowledgebase [30] is one of the most popular databases. It is an open-access, manually curated, and peer-reviewed signaling pathway database. It is used for visualization, interpretation, and analysis of signaling pathways to support basic and clinical research. Our AAAKB aims to provide a similar curated database to meet the need of researchers. Another great example is the m6A-Altas [31], a comprehensive knowledgebase for unraveling the

N6-methyladenosine (m6A) epitranscriptome. According to the help of m6A-Atlas, DICER1-AS1 was identified as a potential target of YTHDF3 with the m6A site in the exon region [32]. All these powerful tools equip researchers with the ability of predicting essential molecules in biological processes. Herein, AAAKB also provides the prediction service for users with high accuracy.

Here, the first comprehensive database (AAAKB), based on the manual curation of AAA-related genes from the literature, has been developed. As a research hotspot, new genes have been found correlating to the incidence of AAA every year. Keeping up with all this information is a daunting task for most researchers. AAAKB helps them in the following ways: 1) It provides up-to-date, comprehensive genetic information relevant to AAA; 2) It contributes to scientific discovery by collecting, interpreting and organizing information so that it is easily accessible and usable; 3) It integrates tools to visualize interactions between specific proteins. This allows for a more intuitive and rapid search for the shortest interaction paths between proteins, contributing to the understanding of the molecular mechanisms of AAA; 4) It provides links to relevant information in other biological databases to help researchers access additional information in more specialized collections.

In our database, we provide not only information of known genes in AAA, but also predict genes that function in AAA via the machine learning algorithm. A previous study has predicted abdominal aortic aneurysm target genes by level-2 protein-protein interaction [33] compiled from the Online Mendelian Inheritance in Man database [34] (OMIM). On the contrary, machine learning is a better choice to predict candidate genes, which has become popular in the applications of biological studies. In addition, compared to thoracic aortic aneurysms, about 20% of which are associated with an autosomal inheritance pattern of single gene mutations, AAA appears to present as a polygenic disease with weaker variants [35]. To adjust for AAA specifically, we have selected known AAA-related genes from the literature as the basis for our prediction in addition to the OMIM database. Therefore, our database predicts AAA-corresponding genes with higher precision.

Abdominal aortic aneurysm is a globally concerning topic, and the number of publications related to AAA has been steadily increasing [36–38]. Over the pastdecade, a total of 8,094 publications have been retrieved, including 7,051 articles and 1,043 comments. Sorting out AAA-related genes from such a large body of literature is a tedious task, but important, as these genes serve as potential therapeutic targets for AAA treatment. The collection of genetic information can accelerate the process of screening and verifying candidate molecules, thereby improving the efficiency and success rate of drug development [39–41]. We have also made efforts to discover new therapeutic targets, and machine learning techniques have been employed to predict new genes related to AAA based on the information we have collected. Furthermore, our integrated protein interaction visualization tool provides a fast and convenient way to understand the connections between targets to enable synergistic or counteract effects of complex signaling networks in AAA. In summary, AAAKB is the first professional gene database for studying AAA, and it provides a helpful resource for researchers to gain new insights into AAA.

## Supporting information

**S1 Fig. Venn diagrams with the intersection of genes in AAAKB and TRRUST.** (A, B) The intersections showed that 27 human genes and 28 mouse genes in AAAKB were curated as transcriptional factors (TFs) in the TRRUST database, respectively.
(TIF)

**S1 Table. Differentially expressed genes in AAA patients and mouse models compared to normal controls.**
(XLSX)

## Author Contributions

**Data curation:** Wangyang Tang, Xingyu Xu.

**Project administration:** Di Yao.

**Visualization:** Shuyuan Mei.

**Writing – original draft:** Zhiguang Shi.

**Writing – review & editing:** Qiulun Lu.

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
