## [Decision Letter · Decision Letter 0]

6 Apr 2023

PONE-D-23-04419AAAKB: A Manually Curated Database for Tracking and Predicting Genes of Abdominal Aortic Aneurysm (AAA)PLOS ONE

Dear Dr. Shi,

Thank you for submitting your manuscript to PLOS ONE. After careful consideration, we feel that it has merit but does not fully meet PLOS ONE’s publication criteria as it currently stands. Therefore, we invite you to submit a revised version of the manuscript that addresses the points raised during the review process.

We look forward to receiving your revised manuscript.

Kind regards,

Vincenzo Bonnici, PhD

Academic Editor

PLOS ONE

Journal Requirements:

Reviewers' comments:

Reviewer's Responses to Questions

**Comments to the Author**

1. Is the manuscript technically sound, and do the data support the conclusions?

Reviewer #1: Partly

2. Has the statistical analysis been performed appropriately and rigorously? 

Reviewer #1: No

3. Have the authors made all data underlying the findings in their manuscript fully available?

Reviewer #1: Yes

4. Is the manuscript presented in an intelligible fashion and written in standard English?

Reviewer #1: Yes

5. Review Comments to the Author

Reviewer #1: In this study, the authors have developed "AAAKB", a manually curated knowledge base containing genes, SNPs, and pathways associated with AAA. AAAKB also provides predicted genes potentially associated with AAA, based on the protein interaction information of genes collected in the database and using the Support Vector Machine (SVM) algorithm to build the predictive model. The manuscript has good English language usage and a well-organized structure but some points require attention to enhance the quality of the paper:

- Improve the introductory section of the manuscript by highlighting and listing the “contributions” points of this research and the work's limitations;

- The study uses the SVM algorithm to predict potentially AAA-associated genes;

- It is crucial to validate these predictions through more experimental studies, such as gene expression analysis, to confirm their roles in AAA development. Moreover, it is crucial to validate with other ML techniques for comparison (RF,NB,KNN,etc), and in the meantime, describe hyper parameter tuning and settings

- The conclusion section needs a clear exposition on how the initial goal of this study was achieved. I suggest this part to support these with adequate numerical results and references, offering a more critical/discursive view of future research.

6. PLOS authors have the option to publish the peer review history of their article (what does this mean?). If published, this will include your full peer review and any attached files.

Reviewer #1: **Yes: **Giovanni Cicceri

---

## [Author Response · Author response to Decision Letter 0]

12 May 2023

Response to Reviewers Comments

We thank the reviewer for his efforts and useful comments that helped us clarify and improve our study and manuscript. We have addressed each comment below and made accompanying changes to the revised manuscript.

- Improve the introductory section of the manuscript by highlighting and listing the “contributions” points of this research and the work's limitations;

Response: Thank you for your valuable comments. We have revised the introductory section of the manuscript to highlight the contributions and limitations of our research on Page 3-4.

- The study uses the SVM algorithm to predict potentially AAA-associated genes; It is crucial to validate these predictions through more experimental studies, such as gene expression analysis, to confirm their roles in AAA development. Moreover, it is crucial to validate with other ML techniques for comparison (RF,NB,KNN,etc), and in the meantime, describe hyper parameter tuning and settings.

Response: Good point. Following your suggestion, I involved another three ML techniques (RF, NB, KNN) for prediction. And you are right that RF overall performs the best. Therefore, we chose RF as the optimal and final prediction model for predicting potentially AAA-associated genes. We have updated this section of the results on page 8.

Regarding the hyperparameter tuning and settings, we apologize for not providing these details in the original manuscript. In the revised manuscript, we have added sections in methods detailing the ML model parameters used for gene prediction on page 4-5.

We agree that it is important to validate the ML-based predictions through gene expression analysis to confirm their roles in AAA development. In our revised manuscript, we have emphasized this point in the result section on page 8, and we have updated the corresponding method on page 5.

- The conclusion section needs a clear exposition on how the initial goal of this study was achieved. I suggest this part to support these with adequate numerical results and references, offering a more critical/discursive view of future research.

Response: Thank you for your suggestion. We have revised the conclusion section to better highlight the contributions of our research and the importance of our gene database in advancing AAA research on page 10-11. In this version, we have discuss the potential of our predicted AAA genes as therapeutic targets, and how it could facilitate researchers to better select their focused gene in the sophisticated signaling network.

---

## [Decision Letter · Decision Letter 1]

31 Jul 2023

AAAKB: A Manually Curated Database for Tracking and Predicting Genes of Abdominal Aortic Aneurysm (AAA)

PONE-D-23-04419R1

Dear Dr. Shi,

We’re pleased to inform you that your manuscript has been judged scientifically suitable for publication and will be formally accepted for publication once it meets all outstanding technical requirements.

Kind regards,

Vincenzo Bonnici, PhD

Academic Editor

PLOS ONE

Additional Editor Comments (optional):

Reviewers' comments:

Reviewer's Responses to Questions

**Comments to the Author**

1. If the authors have adequately addressed your comments raised in a previous round of review and you feel that this manuscript is now acceptable for publication, you may indicate that here to bypass the “Comments to the Author” section, enter your conflict of interest statement in the “Confidential to Editor” section, and submit your "Accept" recommendation.

Reviewer #2: All comments have been addressed

2. Is the manuscript technically sound, and do the data support the conclusions?

Reviewer #2: Yes

3. Has the statistical analysis been performed appropriately and rigorously? 

Reviewer #2: Yes

4. Have the authors made all data underlying the findings in their manuscript fully available?

Reviewer #2: Yes

5. Is the manuscript presented in an intelligible fashion and written in standard English?

Reviewer #2: Yes

6. Review Comments to the Author

Reviewer #2: In the new version of the manuscript, the language use and the structure organization have been greatly improved, making it more reader-friendly and accessible. Overall, quality of the paper was improved and all essential aspects for reading and reproducibility of the work have been added.

7. PLOS authors have the option to publish the peer review history of their article (what does this mean?). If published, this will include your full peer review and any attached files.

Reviewer #2: **Yes: **Giovanni Cicceri

---

## [Editor Report · Acceptance letter]

2 Aug 2023

PONE-D-23-04419R1 

AAAKB: A Manually Curated Database for Tracking and Predicting Genes of Abdominal Aortic Aneurysm (AAA) 

Dear Dr. Shi:

I'm pleased to inform you that your manuscript has been deemed suitable for publication in PLOS ONE. Congratulations! Your manuscript is now with our production department. 

Kind regards, 

on behalf of

Dr. Vincenzo Bonnici 

Academic Editor

PLOS ONE